# Data center cooling using model-predictive control

**Nevena Lazic, Tyler Lu, Craig Boutilier, Moonkyung Ryu**
Google Research
{nevena, tylerlu, cboutilier, mkryu}@google.com

**Eehern Wong, Binz Roy, Greg Imwalle**
Google Cloud
{ejwong, binzroy, gregi}@google.com

## Abstract

Despite the impressive recent advances in reinforcement learning (RL) algorithms, their deployment to real-world physical systems is often complicated by unexpected events, limited data, and the potential for expensive failures. In this paper, we describe an application of RL "in the wild" to the task of regulating temperatures and airflow inside a large-scale data center (DC). Adopting a data-driven, model-based approach, we demonstrate that an RL agent with little prior knowledge is able to effectively and safely regulate conditions on a server floor after just a few hours of exploration, while improving operational efficiency relative to existing PID controllers.

## 1   Introduction

Recent years have seen considerable research advances in reinforcement learning (RL), with algorithms achieving impressive performance on game playing and simple robotic tasks [24, 29, 27]. However, applying RL to the control of real-world physical systems is complicated by unexpected events, safety constraints, limited observations and the potential for expensive or even catastrophic failures. In this paper, we describe an application of RL to the task of data center (DC) cooling. DC cooling is a test bed that is well-suited for RL deployment because it involves control of a complex, large-scale dynamical system, non-trivial safety constraints and the potential for considerable improvements in energy efficiency.

Cooling is a critical part of DC infrastructure, since multiple servers operating in close proximity produce a considerable amount of heat and high temperatures may lead to lower IT performance or equipment damage. There has been considerable progress in improving cooling efficiency, and best-practice physical designs are now commonplace in large-scale DCs [7]. However, on the software side, designing resource-efficient control strategies is still quite challenging, due to complex interactions between multiple non-linear mechanical and electrical systems. Most existing controllers tend to be fairly simple, somewhat conservative, and hand-tuned to specific equipment architectures, layouts, and configurations. This leaves potential for efficiency improvement and automation using more adaptive, data-driven techniques.

As the number of DCs increases with the adoption of cloud-based services, data growth, and hardware affordability, power management is becoming an important challenge in scaling up. In 2014, DCs accounted for about 1.8% of the total power usage in the U.S. and about 626 billion liters of water were consumed by DC operations [28]. There has been increased pressure to improve operational efficiency due to rising energy costs and environmental concerns. This includes cooling, which constitutes a non-trivial part of the DC power overhead.

Recently, DeepMind demonstrated that it is possible to improve DC power usage efficiency (PUE) using a machine learning approach [13]. In particular, they developed a predictive model of PUE in a large-scale Google DC, and demonstrated that it can be improved by manipulating the temperature of the water leaving the cooling tower and chilled water injection setpoints. In this work, we focus on a complementary aspect of DC cooling: regulating the temperature and airflow inside server floors by controlling fan speeds and water flow within air handling units (AHUs).

Our approach to cooling relies on *model-predictive control (MPC)*. Specifically, we learn a linear model of the DC dynamics using safe, random exploration, starting with little or no prior knowledge. We subsequently recommend control actions at each time point by optimizing the cost of model-predicted trajectories. Rather than executing entire trajectories, we re-optimize at each time step. The resulting system is simple to deploy, as it does not require historical data or a physics-based model. The main contribution of the paper is to demonstrate that a controller relying on a coarse-grained linear dynamics model can safely, effectively, and cost-efficiently regulate conditions in a large-scale commercial data center, after just a few hours of learning and with minimal prior knowledge. By contrast, characterizing and configuring the cooling control of a new data center floor typically takes weeks of setup and testing using existing techniques.

## 2 Background and related work

Among approaches in the literature, the most relevant to our problem is *linear quadratic (LQ) control*. Here it is assumed that system dynamics are linear and the cost is a quadratic function of states and controls. When the dynamics are known, the optimal policy is given by constant linear state feedback and can be solved efficiently using dynamic programming. In the case of unknown dynamics, *open-loop* strategies *identify* the system (i.e., learn the parameters of a dynamics model) in a dedicated exploration phase, while *closed-loop* strategies control from the outset, updating models along the way [20].

The simplest closed-loop approach, known as *certainty equivalence*, updates the parameters of the dynamics model at each step and applies the control law as if the estimated model were the ground truth. This strategy is unable to identify the system in general: parameters may not converge, or may converge to the wrong model, leading to strictly suboptimal control [6]. More recent approaches [8, 2, 17] use *optimism in the face of uncertainty*, where at each iteration the algorithm selects the dynamics with lowest attainable cost from some confidence set. While optimistic control is asymptotically optimal [8] and has a finite-time regret bound of $O(\sqrt{T})$ [2], it is highly impractical as finding the lowest-cost dynamics is computationally intractable. Similar regret bounds can be derived using Thompson sampling in place of optimization [3, 4, 25], but most of these approaches make unrealistic stability assumptions about the intermediate controllers, and can in practice induce diverging state trajectories in early stages.

In the open-loop setting, critical issues include the design of exploratory inputs and estimation error analysis. Asymptotic results in linear system identification (see [21]) include one simple requirement on the control sequence, *persistence of excitation* [5]. A review of frequency-domain identification methods is given in [10], while identification of auto-regressive time series models is covered in [9]. Non-asymptotic results are limited and often require additional stability assumptions [16, 14]; most recently, Dean et al. [11] have related the estimation error to the smallest eigenvalue of the finite-time controllability Gramian.

In the presence of constraints on controls or states, the optimal LQ controller is no longer given by linear feedback, and it is usually simpler to directly optimize control variables. In *model-predictive control*, the controller generates actions at each step by optimizing the cost of a model-predicted trajectory. Re-optimizing at each time step mitigates the impact of model error and unexpected disturbances at the expense of additional computation. MPC has previously been used to regulate building cooling [18, 22, 23, 13, 12], with most approaches relying on historical data and physics-based models. In the context of DC cooling, MPC has been used to control adaptive-vent floor tiles in addition to air-conditioning units, with system identification performed via random exploration [30]. In this work, we develop a similar control strategy that relies on open-loop linear system identification, followed by MPC. We demonstrate that our system can successfully control temperatures and airflow in a large-scale DC after only a few hours of safe, randomized exploration.

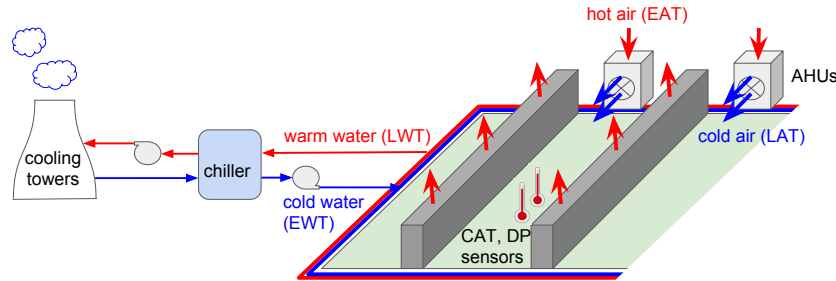

Figure 1: Data center cooling loop. AHUs on the server floor regulate the air temperature through air-water heat exchange. Warmed water is cooled in the chiller and evaporative cooling towers.

## 3   Data center cooling

Figure 1 shows a schematic of the cooling loop of a typical DC. Water is cooled to sub-ambient temperatures in the chiller and evaporative cooling towers, and then sent to multiple air handling units (AHUs) on the server floor. Server racks are arranged in rows between alternating hot and cold aisles. All hot air exhausts into the adjacent hot aisles, which are typically isolated using a physical barrier to prevent hot and cold air from mixing. The AHUs circulate air through the building; hot air is cooled through air-water heat exchange in the AHUs, and blown into the cold aisle. The generated warm water is sent back to the chiller and cooling towers. Naturally, variations of this setup exist.

Our focus is on floor-level cooling, where the primary goal is to regulate cold-aisle temperatures and differential air pressures. Controlling the cold-aisle temperatures ensures that the machines operate at optimal efficiency and prevents equipment damage. Maintaining negative differential air pressure between adjacent hot and cold aisles ensures that cool air flows over servers and improves power efficiency by minimizing the need for the servers to use their own fans. Our goal is to operate close to (but not exceeding) upper bounds on temperature and pressure at minimal AHU power and water usage. Variables relevant to this problem are continuous-valued, and can be grouped as follows:

- *Controls* are the variables we can manipulate. These are *fan speed* (controlling air flow) and *valve opening* (which regulates amount of water used) for each AHU.

- *States* collect the process variables we wish to predict and regulate. These include *differential air pressure (DP)* and *cold-aisle temperature (CAT)*, measured using multiple sensors along the server racks. To reduce redundancy and increase robustness to failed sensors, we model and regulate the median values of local groups of CAT and DP sensors. We also measure the *entering air temperature (EAT)* of the hot air entering each AHU, and *leaving air temperature (LAT)* of the cooled air leaving each AHU.

- *Disturbances* are the events and conditions which we cannot manipulate or control, but which nonetheless affect the conditions inside the server floor. These include *server power usage*, which serves as a proxy for the amount of generated heat, as well as the *entering water temperature (EWT)* of the chilled water measured at each AHU.

An illustrative schematic of the structure of the DC used in our case study is shown in Figure 1. The system consists of many dozens of AHUs, with two controls each, and many dozens of state variables *for each row*. The existing cooling system relies on local PID controllers (one per AHU), which are manually tuned and regulate DP measured at nearby sensors and LAT measured at the same AHU. Directly controlling CAT (the variable of interest) instead of LAT is more complicated, as temperatures along the server racks take a longer time to respond to changes in controls and depend on multiple AHUs. Since the local controllers operate independently, they may settle into a suboptimal state where some AHUs do little work while others run at their maximum capacity to compensate. This is addressed using a supervisory software layer which heuristically readjusts local controls to operate in a more balanced state.

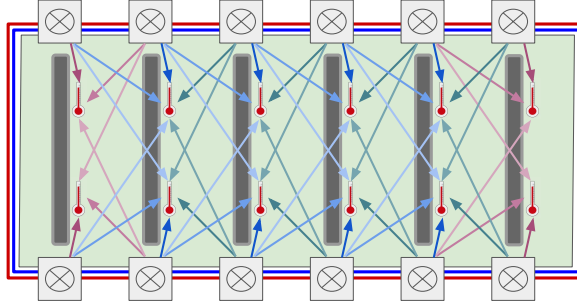

Figure 2: Model structure illustration. Sensor measurements at each location only depend on the closest AHUs. The regularity of DC layout allows parameters to be shared between local models with the same structure (arrows with the same color share weights).

Table 1: State variable dependencies

| Variable | Predictors |
|----------|------------|
| DP | DP measurements and fan speeds in up to 5 closest aisles / 10 closest AHUs |
| LAT | LAT, EWT, EAT, fan speed, and valve position at the closest AHU |
| CAT | CAT, LAT, and fan speeds in up to 3 closest aisles / 6 closest AHUs |
| EAT | EAT, CAT, fan speeds, and power usage at up to 3 closest aisles / 6 closest AHUs |

## 4 Model predictive control

We consider the use of MPC to remove some of the inefficiencies associated with the existing PID control system. We: (i) model the effect of each AHU on state variables in a large neighborhood (up to 5 server rows) rather than on just the closest sensors; (ii) control CAT directly rather using LAT as a proxy; and (iii) jointly optimize all controls instead of using independent local controllers. We identify a model of DC cooling dynamics using only a few hours of exploration and minimal prior knowledge. We then control using this learned model, removing the need for manual tuning. As we show, these changes allow us to operate at less conservative setpoints and improve the cooling operational efficiency.

### 4.1 Model structure

Let $\mathbf{x}[t], \mathbf{u}[t]$, and $\mathbf{d}[t]$ be the vectors of all state, control, and disturbance variables at time $t$, respectively. We model data center dynamics using a linear auto-regressive model with exogeneous variables (or ARX model) of the following form:

$$\mathbf{x}[t] = \sum_{k=1}^{T} A_k \mathbf{x}[t-k] + \sum_{k=1}^{T} B_k \mathbf{u}[t-k] + C\mathbf{d}[t-1] \,. \tag{1}$$

where $A_k$, $B_k$, and $C_k$ are parameter matrices of appropriate dimensions. Since we treat sensor observations as state variables, our model is $T$-Markov to capture relevant history and underlying latent state. Each time step corresponds to a period of 30s, and we set $T = 5$ based on cross-validation. While the true dynamics are not linear, we will see that a linear approximation in the relevant region of state-action space suffices for effective control.

We use prior knowledge about the DC layout to impose a block diagonal-like sparsity on the learned parameter matrices. The large size of the server floor allows us to assume that temperatures and DPs at each location directly depend only states, controls, and disturbances at nearby locations (i.e., are conditionally independent of more distant sensors and AHUs given the nearby values).[1] Additional parameter sparsity can be imposed based on variable types; for example, we know that DP directly

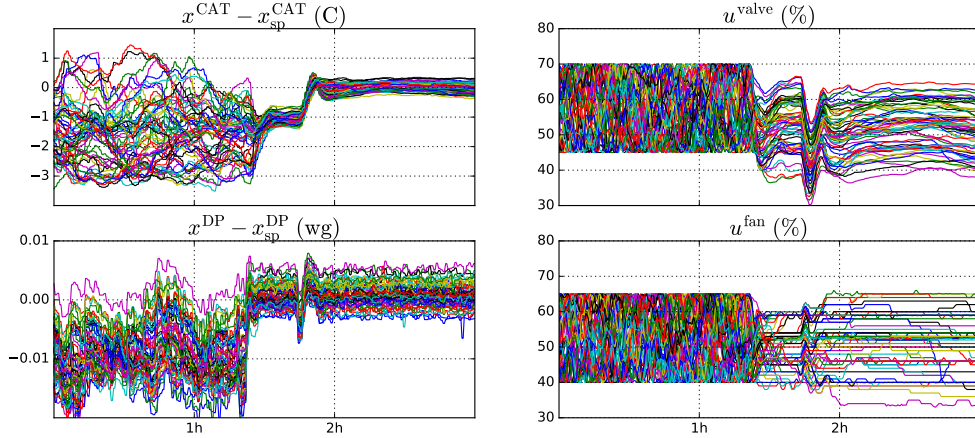

Figure 3: An example run of random exploration, followed by control. The figure shows valve commands and fan speeds for all AHUs, as well as the CAT and DP sensor values at multiple locations throughout the DC. The system controls DP at a setpoint $x_{\text{sp}}^{\text{DP}}$. CAT control starts at setpoint $x_{\text{sp}}^{\text{CAT}} - 1$ followed by $x_{\text{sp}}^{\text{CAT}}$; the temperatures transition between the two values quickly and with little overshoot.

depends on the fan speeds, but is (roughly) independent of temperature within a narrow temperature range. We list the features used to predict each state variable in Table 1.

Since the servers, sensors, and cooling hardware are arranged in a regular physical layout in the DC we work with, we share parameters between local models for sample efficiency. Thus, our model has an overall linear convolutional structure, as illustrated in Figure 2.

## 4.2  System identification

We learn the system dynamics using randomized exploration over controls, starting with a "vacuous" model that predicts no change in states. While we have access to historical data generated by the local PID controllers, it is not sufficiently rich to allow for system identification due to the steady state behaviour of the controllers.[2] During the control phase we continue to update the dynamics in an online or batch-online fashion.

As safe operation during exploration is critical, we limit each control variable to a safe range informed by historical data. In the absence of such data, the safe range can be initialized conservatively and gradually expanded. We also limit the maximum absolute changes in fan and valve controls between consecutive time steps since large changes may degrade hardware over time. Let $u_i^c[t]$ indicate the value of the control variable $c$ for the $i^{th}$ AHU at time step $t$, with $c \in \{\text{fan}, \text{valve}\}$. Let $[u_{\min}^c, u_{\max}^c]$ be the range of control variable $c$, and let $\Delta^c$ be the maximum allowed absolute change in $c$ between consecutive time steps. Our exploration strategy is a simple, range-limited uniform random walk in each control variable:

$$u_i^c[t+1] = \max(u_{\min}^c, \min(u_{\max}^c, u_i^c[t] + v_i^c)), \quad v_i^c \sim \text{Uniform}(-\Delta^c, \Delta^c). \quad (2)$$

This strategy ensures sufficient frequency content for system identification and respects safety and hardware constraints. Figure 3 shows controls and states during an example run of random exploration, followed by control.

During the exploration phase, we update model parameters using recursive least squares [15]. In the control phase, we update parameters selectively so as not to overwhelm the model with steady-state data. In particular, we estimate the noise standard deviation $\sigma_s$ for each variable $s$ as the root mean squared error on the training data, and update the model with an example only if its (current) prediction error exceeds $2\sigma_s$.[3]

## 4.3 Control

Given our model and an initial condition (the $T$ past states, controls, and disturbances for the $M$ AHUs), we optimize the cost of a length-$L$ trajectory with respect to control inputs at every step. Let $x_{\mathrm{sp}}^s$ denote the setpoint (or target value) for a state variable $s$, where $s \in \{\mathrm{DP, CAT, LAT}\}$. Let $x_i^s[t]$ denote the value of the state variable $s$ for the $i^{th}$ AHU at time $t$. We set controls by solving the following optimization problem:

$$\min_{\mathbf{u}} \sum_{\tau=t}^{t+L} \sum_{i=1}^{M} \left( \sum_s q_s (x_i^s[\tau] - x_{\mathrm{sp}}^s)^2 + \sum_c r_c (u_i^c[\tau] - u_{\mathrm{min}}^c)^2 \right) \qquad (3)$$

$$\text{s.t. } u_i^c \in [u_{\mathrm{min}}^c, u_{\mathrm{max}}^c], \ \ |u_i^c[\tau] - u_i^c[\tau-1]| \le \Delta^c, \ \ \mathbf{d}[\tau] = \mathbf{d}[\tau-1] \qquad (4)$$

$$\mathbf{x}[\tau] = \sum_{k=1}^{T} A_k \mathbf{x}[\tau-k] + \sum_{k=1}^{T} B_k \mathbf{u}[\tau-k] + C\mathbf{d}[\tau-1] \qquad (5)$$

$$t \le \tau \le t+L, \ c \in \{\mathrm{fan, valve}\}, \ s \in \{\mathrm{DP, CAT, LAT}\}. \qquad (6)$$

Here $q_s$ and $r_c$ are the weights for the loss w.r.t. state and control variables $s$ and $c$, respectively, and $i$ ranges over AHUs. We assume that disturbances do not change over time. Overall, we have a quadratic optimization objective in $2ML \simeq 1.2K$ variables, with a large number of linear and range constraints. While we optimize over the entire trajectory, we only execute the optimized control action at the first time step. Re-optimizing at each step enables us to react to changes in disturbances and compensate for model error.

We specify the above objective as a computation graph in TensorFlow [1] and optimize controls using the Adam [19] algorithm. In particular, we implement constraints by specifying controls as

$$u_i^c[\tau] = \max(u_{\mathrm{min}}^c, \min(u_{\mathrm{max}}^c, u_i^c[\tau-1] + \Delta^c \tanh(z_i^c[\tau]))) \qquad (7)$$

where $z_i^c[\tau]$ is an unconstrained optimization variable. The main motivation for this choice is its simplicity and speed—the optimization converges well before our re-optimization period of 30s.

## 5 Experiments

We evaluate the performance of our MPC approach w.r.t. the existing local PID method on a large-scale DC. Since the quality of MPC depends critically on the quality of the underlying model, we first compare our system identification strategy to two simple alternatives. One complication in comparing the performance of different methods on a physical system is the inability to control environmental disturbances which affect the achievable costs and steady-state behavior. In our DC cooling setting, the main disturbances are the EWT (temperature of *entering* cold water) and server power usage (a proxy for generated heat). These variables also reflect other factors (e.g., weather, time of day, server hardware, IT load). To facilitate a meaningful comparison, we evaluate the cost of control (i.e., cost of power and water used by the AHUs) for different ranges of states and disturbances.

### 5.1 System identification

We first evaluated our system identification strategy by comparing the following three models:

- Model 1: our model, trained on 3 hours of deliberate exploration data with controls following independent random walks limited to a safe range as described in Section 4.2.
- Model 2: trained on a week of historical data generated by local PID controllers. While this model is trained on 56 times more data than the others, it turns out that the data is not as informative. Since local controllers regulate LAT to a fixed offset above EWT, the model may simply learn this relationship rather than the dependence of LAT on controls. Furthermore, if state values do not vary much, it may learn to predict no changes in state.
- Model 3: trained on 3 hours of data with controls recommended by a certainty-equivalent controller (i.e., optimal controls w.r.t. all available data at each iteration, see Section 2), limited to a safe range. While this data contains a wider range of inputs than Model 2 data, it contains no exploratory actions.

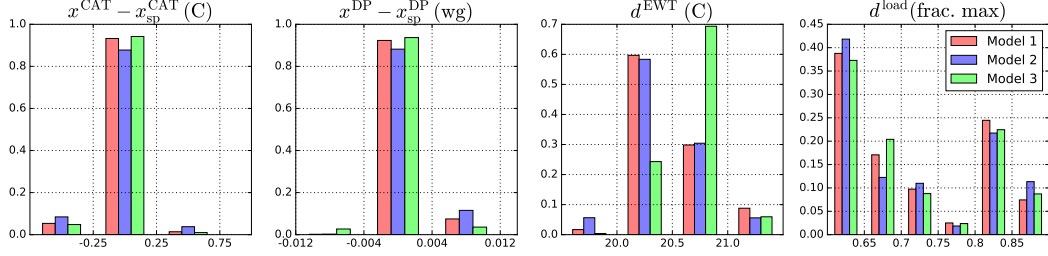

Figure 4: Histograms of state variables and disturbances over time and AHUs during steady-state operation of MPC controllers using three different models.

Table 2: Average power and water cost (% data) for each controller, restricted to time points and AHUs for which CAT was within $0.25C$ of $x_{\text{sp}}^{\text{CAT}}$ and pressure within $0.004$ of $x_{\text{sp}}^{\text{DP}}$, stratified by values of the disturbances.

| Entering water temperature (C) | Server load (fraction of max) | Model 1 cost (% data) | Model 2 cost (% data) | Model 3 cost (% data) |
|---|---|---|---|---|
| $\leq 20.5$ | $\leq 0.7$ | **84.3** (31%) | 94.4 (29.9%) | 99.6 (13.7%) |
| $> 20.5$ | $\leq 0.7$ | **85.8** (17.6 %) | 93.8 (14.1 %) | 112.7 (36.0 %) |
| $\leq 20.5$ | $> 0.7$ | **142.4** (21.9 %) | 149.4 (20.4 %) | 178.2 (8.3 %) |
| $> 20.5$ | $> 0.7$ | **144.6** (15.3 %) | 148.9 (12.8 %) | 182 .1 (29.9 %) |
| any | any | **110.2** (85.8%) | 117.9 (77.2%) | 140.4 (87.9%) |

We controlled median CAT and DP at setpoints $x_{\text{sp}}^{\text{CAT}}$ and $x_{\text{sp}}^{\text{DP}}$, using each model for approximately one day. We examine the steady state behavior of the controllers next. Figure 4 shows histograms of states and disturbances during the operation of the three controllers, with data aggregated over both time and sensors. In all three cases, state variables remain close to their targets most of the time, but the controller based on Model 2 (historical data) had the highest steady-state error (e.g., the difference between CAT/DP and their setpoints is close to zero less often with Model 2). The distribution of server loads during the three tests was similar, while the EWT was somewhat higher for Model 3. The average cost of controls (fan power and water usage in the objective) was 115.9, 116.6, and 139.9, respectively; however, these are not directly comparable due to differences in steady state error and disturbances.

Stratifying data by state and disturbance values is somewhat complicated. For example, sensor measurements at any location are affected by multiple AHUs with different EWTs. Similarly, each AHU affects measurements at multiple racks with different loads. To simplify analysis, we treat each group of sensors as dependent on its closest AHU, allowing independent consideration of each AHU. A lesser complication is the time lag between control changes and state changes. Since controllers largely operate in steady state, controls do not change often, so we consider time points independently.

To compare costs, we first restrict available data to time points and AHUs where temperatures were within $0.25C$ of $x_{\text{sp}}^{\text{CAT}}$, and pressures within $0.004$ of $x_{\text{sp}}^{\text{DP}}$ (i.e., the intersection of histogram peaks in Figure 4, left). This corresponded to 85.8%, 77.2%, and 87.9% of the data for controllers using Models 1, 2, and 3, respectively. We then stratified the data by different ranges of EWT and server load. We evaluated the control cost for each disturbance range. The results are shown in Table 2, and suggest that the controller based on Model 1 (with explicit exploration data) is the most efficient.

## 5.2 Comparison to local PID controllers

The existing local PID controllers differ from ours in that they regulate LAT to a constant offset relative to EWT, rather than controlling CAT directly. To compare the two approaches, we ran our MPC controller with the same LAT-offset setpoints for one day, and compared it to a week of local PID control. As before, we treat measurements at each group of sensors as depending only on the closest AHU, and ignore time lags (assuming reasonable control consistency during steady-state operation). Histograms of states and disturbances during the operation of the two controllers are

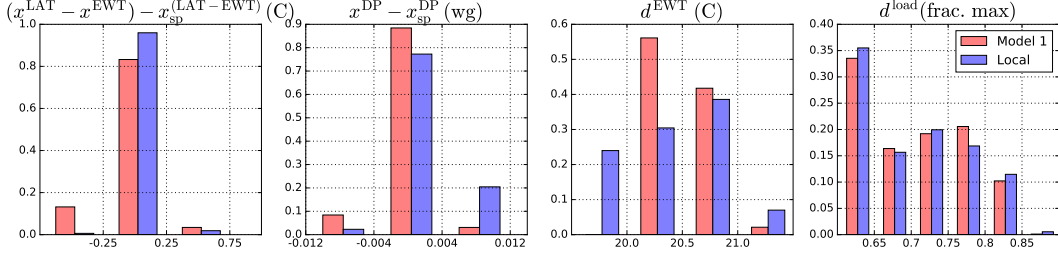

Figure 5: Histograms of state variables and disturbances over time and AHUs during steady-state operation of the MPC (Model 1) and local PID controllers.

Table 3: Average total cost (% data) of each controller, restricted to time points and fancoils for which LAT-EWT and DP were within 0.25C and 0.004wg of their respective setpoints, stratified by values of the disturbances.

| Entering water temperature (C) | Server load (frac. max) | Local controllers cost (% data) | Model 1 cost (% data) |
|---|---|---|---|
| $\leq 20.5$ | $\leq 0.7$ | **95.3** (19.8 %) | 106.4 (22.6 %) |
| $> 20.5$ | $\leq 0.7$ | 107.9 (13.8 %) | **104.9** (15.0 %) |
| $\leq 20.5$ | $> 0.7$ | 170.3 (20.1%) | **130.5** (18.8 %) |
| $> 20.5$ | $> 0.7$ | 187.8 (20.4 %) | **128.7** (18.0 %) |
| any | any | 142.2 (74.4%) | **116.7** (74.1%) |

shown in Figure 5. Local controllers track the temperature setpoint more closely, but operate at higher DP. Server loads are similar, while average EWT is lower during local controller operation.

To compare costs, we restrict data to AHUs and times corresponding to the peaks of histograms in Figure 5 left (about 74% of the data for both controllers). We stratify this data as above and compare the total cost in each stratum in Table 3. While local control was more cost efficient under low EWT and server load, our controller was more efficient under all other conditions and overall.

While the quadratic objective is a reasonable approximation, it does not correspond exactly to the true dollar cost of control, which is not quadratic and may change over time. After restricting to temperatures and pressures as in Tables 3 and 2, the average dollar cost (units unspecified) of our LAT and CAT controllers was 94% and 90.7% of the cost of the local controllers. While precise quantification of these savings requires longer-term experiments, our approach of jointly optimizing controls of all AHUs, together with the ability to control process variables at slightly higher values, has the potential to save about 9% of the current server-floor cooling costs.

## 6 Discussion

We have presented an application of model-based reinforcement learning to the task of regulating data center cooling. Specifically, we have demonstrated that a simple linear model identified from only a few hours of exploration suffices for effective regulation of temperatures and airflow on a large-scale server floor. We have also shown that this approach is more cost effective than commonly used local controllers and controllers based on non-exploratory data.

One interesting question is whether the controller performance could further be improved by using a higher-capacity model such as a neural network. However, such a model would likely require more than a few hours of exploratory data to identify, and may be more complicated to plan with. Perhaps the most promising direction for model improvement is to learn a mixture of linear models that could approximate dynamics better under different disturbance conditions. In terms of overall data center operational efficiency, further advantages are almost certainly achievable achieved by jointly controlling AHUs and the range of disturbance variables if possible, or by planning AHU control according to known/predicted disturbances values rather than treating them as noise.

**Acknowledgments**

The experiments performed for this paper would not have been possible without the help of many people. We would especially like to thank Dave Barker, Charisis Brouzioutis, Branden Davis, Orion Fox, Daniel Fuenffinger, Amanda Gunckle, Brian Kim, Eddie Pettis, Craig Porter, Dustin Reishus, Frank Rusch, Andy Thompson, and Abbi Ward. We also thank Gal Elidan for many valuable discussions.

## Footnotes

[1]In other words, the nearby sensors and controls form a Markov blanket [26] for specific variables in a graphical model of the dynamical system.

[2]Specifically, the PID controllers operate in too narrow a range of (joint) state-control space to generate data allowing sufficiently accurate prediction in novel regions.

[3]In long running operation, triggering further exploration to account for rare exogenous conditions or disturbances (as well as drift) may be necessary, but we don't consider this here.

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
