[Reviews · NeurIPS 2018]

Reviewer 1



This paper addresses the problem of temperature and airflow regulation for a large-scale data center and considers how a data-driven, model-based approach using Reinforcement Learning (RL) might improve operational efficiency relative to the existing approach of hand-crafted PID controllers. Existing controllers in large-scale data centers tend to be simple, conservative and hand-tuned to physical equipment layouts and configurations. Safety constraints and a low tolerance for performance degradation and equipment damage impose additional constraints. The authors use model-predictive control (MPC) to learn a linear model of the data center dynamics (a LQ controller) using safe, random exploration, starting with little or no prior knowledge. They then determine the control actions at each time step by optimizing the cost of the model-predicted trajectories, ensuring to re-optimize at each time step. The proposed work is not the first to apply MPC to data center cooling. Zhou et al. (2012) used MPC to control adaptive-vent floor tiles and air-conditioning units with a system identificaition done via random exploration. Here, the authors perform an open-loop linear system identification, then follow it by MPC. Their approach does not depend on historical or physics-based models. In fact, they show that the use of historical data might even be harmful to the control performance since the model learnt on observed data might not capture the key underlying dependencies in the system. This seems obvious, in light of the fact that in a data center, state values are unlikely to vary much due to the steady-state of the PID controllers. Hence, observed data might not be rich enough for system identification. This paper is easy to understand, with the background of data center cooling explained clearly with diagrams and the list of variables (controls, states, disturbances) linked to predictors. I enjoyed reading this application paper since the authors were careful in setting up the context for the relevance and challenges associated with the temperature and airflow control problem. The problem examined in this paper is certainly a relevant one, given the increasing demand for computing resources and the corresponding expectation for performance and reliability. The contribution here is less algorithmic in nature but more on the illustration of a open-loop linear system identification approach + MPC on a real-world large-scale dynamical system. It is an interesting application of RL deployed to a significant problem domain. I have read the authors' response and thank them for their comments.

Reviewer 2



This paper describes a very interesting application of RL to the task of regulating temperatures and airflow inside a large-scale data center. It is demonstrated that the RL agent is able to effectively and safely regulate conditions in the data center after just a few hours of exploration with little prior knowledge. This will lead to a reduction in the hand-tuning effort that goes for data centers which is often specific to the equipment architectures and configurations. This has advantages over the recent work by DeepMind which used machine learning to manipulate the temperature of the water leaving the cooling tower and chilled water injection setpoints. Particularly, this paper uses model-predictive control which learns a linear model of the data center dynamics using random (but always safe) exploration requiring no physics-based model or even any historical data. Then, the controller generates actions at each step by optimizing the cost of a model-predicted trajectory. Key highlights: The novelty and the neatness of the problem definition is striking and this will motivate new research in this application. The paper clearly describes all the assumptions and motivates them from the perspective of this novel application. The paper describes the development and demonstrated effectiveness of the various structural modeling assumptions in this domain which is impressive. Particularly, the method even during exploration is always safe. Some of the main issues: The paper alludes to prior work by DeepMind as close to this work. But this is not described to compared to in the experiments. It would be good to have that comparison or atleast a discussion of why that is not possible. There is some notation that has not been clearly defined in the paper. While that is understandable from context, it will be good to have a table of notation to make it convenient for readers. For example, delta, u_min, u_max are undefined. Some of the model choices can be better motivated. For example, lines 169-171 - "we estimate the noise standard deviation σs for each variable s as the root mean squared error on the training data, and update the model with an example only if its (current) prediction error exceeds 2σs" Why this choice? Some of the writing, especially in section 4.3 can be simplified. Equation 3 is difficult to parse, This is essentially weighted least squares. It will be beneficial to the reader to start with that information, And maybe introduce notation before the equation. Why are x_sp^s and u^c_min used in the difference in the square. Questions: a) What is "target setpoint value"? how's it computed? b) Lines 183-186. What is the reason that the reformulated optimization converges well before the cvxpy implementation? This can be better discussed. I read the author response. Thanks!

Reviewer 3



This paper applies model predictive control (MPC) for data center cooling, i.e. regulating air temperature and air flow on server room/floor by controlling blower speeds and water flow at air handling units (AHUs). The dynamic model for the MPC is assumed to be a linear autoregressive model and its parameters are learned from randomized exploration over controls. The performance of the approach is evaluated by field experiments at a data center (DC), discussed and compared with the classical/traditional PID control. Pros - This is a very practical topic which has great potentials for real-world applications. It is exciting to see the field experiments have been performed (I believe the field experiment is expensive) and the results are promising. - This paper is clearly written, well-organized. The high presentation quality makes this paper an enjoyable read. Cons - The theory/methodology contribution is limited. The paper is more of a demonstration that we can applying existing techniques to a DC for substantial improvements. - I don't like the comparison of computation time with cvxpy in Section 4.3, line 185. The computation time largely depends on programming language, computing hardware, and many details in implementing the optimization library. Besides, given the cooling is a slow process, I'm curious if we can increase our step to be more than 30s. Other Questions/Comments The proposed approach relies on little or no prior knowledge. For an operating DC which has historical recordings, will these historical data somehow help with the model learning, if we want to upgrade its cooling from PID to MPC? I was thinking this because I believe obtaining its historical data is not expensive/difficult. Regarding the CAT/DP sensors. For "failed sensors", could you please add comments on the sensors' failing rates? For "measurement redundancy", how close are the measurement/readings from nearby sensors? I was thinking the adverse impacts of the failed sensor to the control and whether it is beneficial to build a model which contains every sensor's reading. In Fig2, could you please add some explanations/descriptions for the model structure on shared parameters among local models. Is there any symmetry in the structure? I was thinking, (assuming the AHU are located on a 2*6 matrix, zero-based), why AHU_0_0's (top row, leftmost) red arrow points to the right, while AHU_0_5's (top row, rightmost) red arrow also points to the right? Why Column_4 is different from Columns 0/1/2/3 in that its red arrows point to the right? In the exploration stage (Section 4.2), for the safety considerations, are you only limiting the size of control step? Are you also monitoring the temperatures at critical points? In the experiments analysis, the entering water temperature (EWT) is quite different between Model3 and Model1/2. Could you please comment on how will EWT impact the cooling? Is EWT a big impact to cooling efficiency? How difficult is it to make EWT consistent for all experiments, thus that we have a fairer comparison? ======= Phase 2 ========= I have read the response from authors. Thanks for the explanations!